# Blood Eosinophils Matter in Post-COVID-19 Pneumonia

**DOI:** 10.3390/diagnostics14202320

**Published:** 2024-10-18

**Authors:** Nicol Bernardinello, Gioele Castelli, Dylan Pasin, Giulia Grisostomi, Marco Cola, Chiara Giraudo, Elisabetta Cocconcelli, Annamaria Cattelan, Paolo Spagnolo, Elisabetta Balestro

**Affiliations:** 1Respiratory Disease Unit, Department of Cardiac, Thoracic, Vascular Sciences and Public Health, Padova University Hospital, 35128 Padova, Italy; nicol.bernardinello@unipd.it (N.B.); gioele.castelli@phd.unipd.it (G.C.); dylan.pasin@aopd.veneto.it (D.P.); giulia.grisostomi@aopd.veneto.it (G.G.); elisabetta.cocconcelli@aopd.veneto.it (E.C.); paolo.spagnolo@unipd.it (P.S.); 2Infectious Diseases Unit, Department of Internal Medicine, Azienda Ospedaliera Universitaria di Padova, 35128 Padova, Italy; marco.cola@aopd.veneto.it (M.C.); annamaria.cattelan@aopd.veneto.it (A.C.); 3Department of Cardiac, Thoracic, Vascular Sciences, and Public Health, University of Padova, 35128 Padova, Italy; chiara.giraudo@unipd.it

**Keywords:** eosinophils, COVID-19 pneumonia, lung abnormalities, chest CT scan, prognostic markers

## Abstract

Background: Even after the development of vaccines, SARS-CoV-2 continues to cause severe pneumonia all over the world. Consequently, in order to improve the management of patients and optimize the use of resources, predictors of disease severity and lung complications after COVID-19 pneumonia are urgently needed. Blood cell count is an easily available and reproducible biomarker. With this study, we aimed to explore the role of eosinophils in predicting disease behavior and pulmonary sequelae at first follow-up with computed tomography (CT). Methods: we evaluated blood cell count and other inflammatory markers, both at baseline and during hospitalization, in a large population of hospitalized COVID-19 patients. Results: 327 patients were finally enrolled, 214 were classified as low-intensity medical care (LIMC) and 113 as high-intensity medical care. Eosinophils were higher at discharge in the HIMC group [0.1 (0–0.72) vs. 0.05 (0–0.34) × 109/L; *p* < 0.0001]. Moreover, in the multivariable analysis, age ≥ 62 years (OR 1.76 (1.05–2.8) *p* = 0.03) and Δ eosinophils ≥ 0.05 (OR 1.75 (1.05–2.9) *p* = 0.03) were two independent predictors of residual lung abnormalities in the whole patient population at first follow-up. Conclusions: an eosinophil increase during hospitalization could be a potential predictor of pulmonary sequelae in surviving patients after COVID-19 pneumonia.

## 1. Introduction

The SARS-CoV-2 pandemic has become the most critical health emergency of our century. Consequently, predictors of disease complications are needed [1,2,3]. Like other infections, COVID-19 pneumonia is associated with systemic inflammation, and the current literature supports its role in the severity and progression of the disease. White blood cells (WBCs), neutrophil-to-lymphocyte ratio (NLR), eosinophil count, D-Dimer, C-reactive protein (CRP), ferritin, and other inflammatory biomarkers are essential to benchmark the systemic inflammatory response [4,5]. However, the role of these biomarkers in the development and progression of lung sequelae after COVID-19 pneumonia is not fully understood. In the specific case of eosinophils, growing evidence suggests that they can have a possible role as biomarkers for diagnosis, prognosis, and disease course. Of interest, it has been well documented in many studies that eosinophil levels were significantly lower in COVID-19 patients with critical disease compared to those who require lower-intensity care [6,7]. Given this background, our study first investigated a possible association between routine blood tests (especially eosinophils) and the severity of pneumonia during hospitalization. The second goal was to explore the relationship between inflammatory markers and radiological pulmonary sequelae after discharge.

## 2. Materials and Methods

### 2.1. Study Population and Study Design

Patients hospitalized for COVID-19 pneumonia were retrospectively enrolled. The patients were hospitalized between March 2020 and March 2021, during the first and second Italian SARS-CoV-2 waves. According to the local epidemiology, patients were mainly exposed to the SARS-CoV-2 variants alpha and delta [8]. Based on the patients’ clinical conditions during hospitalization, the study population was categorized according to the level of care needed. The use of high-flow nasal cannula, noninvasive ventilation, or invasive mechanical ventilation, which required admission either to the intensive care unit or to the Respiratory intensive care unit, were considered high-intensity medical care (HIMC) settings, while the need for oxygen supplementation through low-flow nasal cannula or face mask was considered a low-intensity medical care (LIMC) setting, as previously described. HIMC and LIMC, as previously shown, can clearly describe the disease course and severity during hospitalization [1]. Clinical, radiological, and demographic data for the population were obtained retrospectively from electronic records at hospital admission and discharge. The most important comorbidities were detected retrospectively and were categorized as cardiovascular diseases, respiratory diseases (like asthma and chronic obstructive pulmonary disease), metabolic diseases (including obesity, dyslipidemia, and diabetes mellitus), autoimmune diseases, and oncological diseases (including lung, prostate, pancreatic, breast, and colon cancer). All patients were radiologically evaluated with a chest CT scan at a three-month follow-up visit. A complete blood test was also collected at admission and discharge from the hospital. ∆ eosinophils were calculated using the same method as reported previously by Chen et al. [5] (∆ eosinophils = eosinophils at discharge minus eosinophils on admission).

### 2.2. Radiological Evaluation

Patients were grouped into Recovered (Rec) and Not-Recovered (Not-Rec) based on the presence of residual lung abnormalities at CT scans during follow-up. Available CT scans at admission (*n* = 80) were also scored, as previously reported [6]. Specifically, ground-glass opacities (GGOs), alveolar score (AS), consolidations (CONSs), and reticulations (interstitial score, IS) were analyzed. The two radiologists assessed the levels of the AS, CONS, and IS using a scale from 0 to 100 and estimated the extent to the nearest 2%. The result was expressed as the mean value of the five lobes in the AS, CONS, and IS. The level of interobserver agreement was obtained for each patient as a mean of 5 lobes, as well as for each radiological abnormality (AS, CONS, and IS), and was expressed as Cohen’s *k* value. Disagreement between radiologists was resolved by consensus.

### 2.3. Statistical Analysis

Mann–Whitney, χ2, and Fisher’s exact test were performed for comparison, as appropriate. The univariable and multivariable logistic regression analysis was applied to evaluate if age, NLR, CRP, D-Dimer, and eosinophils were associated with radiological sequelae at three months. For multivariable analysis, the presence or absence of pulmonary sequelae at three months were dichotomized as “yes”, indicating Recovered (Rec), or “no”, indicating Not Recovered (Not-Rec). Also, for multivariable analysis, all the variables used were dichotomized, considering the median to be the cutoff. A Wilcoxon signed-rank test was used to compare the eosinophil values between admission and discharge. Statistical analysis was performed using SPSS (v. 26, IBM Armonk, NY, USA). The graphics were created using GraphPad Prism software (v. 9, Boston, MA, USA). Data were collected during the first and second pandemic waves, from February 2020 to September 2021. This study was performed following the declaration of Helsinki and was approved by the ethics committee of the University Hospital of Padua (n°46430/03082020).

## 3. Results

In this study, three hundred and twenty-seven (327) patients were enrolled at the University Hospital of Padova (Division of Infectious and Tropical Diseases, Respiratory Diseases Unit, and Intensive Care Unit) and followed at our post-COVID-19 clinic. None of the patients were vaccinated. In the end, one hundred and thirteen (113) patients were classified as HIMC, and two hundred and fourteen (214) were classified as LIMC, as reported in Table 1. SARS-CoV-2 infection was confirmed for all patients enrolled by real-time polymerase chain reaction on nasopharyngeal swabs before or during the hospital stay.

The HIMC group was older [65 (25–88) vs. 60 (22–87) years; *p* = 0.01] and had higher levels of cardiological (62 vs. 43%; *p* = 0.001) and metabolic disorders (58 vs. 40%; *p* = 0.001) compared with the LIMC group. Moreover, at hospital admission, HIMC patients presented a higher rate of deterioration of respiratory gas exchange [FiO2 request: 29 vs. 21%, *p* < 0.0001; PaO2 on room air: 61 vs. 71 mmHg, *p* < 0.0001] compared with LIMC patients. Analyzing the 80 CT scans available at the moment of diagnosis, higher severity scores were reported in the HIMC group for all the scores evaluated [alveolar: 18 vs. 5%; *p* = 0.01; consolidation: 4 vs. 0.8%; *p* = 0.001; and interstitial scores: 10 vs. 0.8%; *p* < 0.0001] in comparison with LIMC. More patients presented with dyspnea in the HIMC group (64 vs. 39%; *p* = 0.0003), whereas anosmia was more frequent in the LIMC group (33 vs. 19%; *p* = 0.009). Inflammatory markers and blood cell count were evaluated at hospitalization and discharge, showing that the HIMC group had a higher WBC count [6.9 (1.63–19.15) vs. 5.46 (1.42–25.6) × 10^9^/L; *p* < 0.0001], neutrophils [5.68 (1.08–19) vs. 3.97 (1.05–22.7) × 10^9^/L; *p* < 0.0001], NLR [7.28 (1.27–93.45) vs. 4.05 (0.73–41) × 10^9^/L; *p* < 0.0001], CRP [98 (3.3–350) vs. 43 (2.9–270) mg/dl; *p* < 0.0001], and ferritin [806 (17–12057) vs. 529 (8.3–4723) ng/mL; *p* < 0.0001]. Conversely, patients in the HIMC group showed a lower eosinophil count [0 (0–0.14) vs. 0 (0–0.24) × 10^9^/L; *p* < 0.0001] and monocyte count [0.38 (0.02–1.71) vs. 0.46 (0.02–1.83) × 10^9^/L; *p* = 0.02]. In contrast to the first blood sample, both lymphocytes [1.84 (0.4–8.71) vs. 1.62 (0.18–4.6) × 10^9^/L; *p* = 0.04] and eosinophils [0.1 (0–0.72) vs. 0.05 (0–0.34) × 10^9^/L; *p* < 0.0001] were higher at discharge in the HIMC group.

We further analyzed the trend in eosinophils from admission to discharge. The low eosinophil count at admission reached higher values in both groups; however, in the HIMC group, eosinophils progressively increased, reaching significantly higher levels than in LIMC patients (0.1 vs. 0.04 × 10^9^/L; *p* < 0.0001) (Figure 1).

Examining the radiological evaluations three months after discharge, we observed a higher percentage of patients in the HIMC group with residual lung abnormalities (Not-Rec group) (64 vs. 41%; *p* = 0.0007). To detect predictors for Not-Rec, a logistic regression was performed. In the univariable analysis, age ≥ 62 years (OR 2.04 (1.3–3.2) *p* = 0.002), HIMC (OR 2.56 (1.6–4.10) *p* = 0.0001), NLR at admission ≥ 4.64 (OR 1.66 (1.07–2.58) *p* = 0.02), neutrophils at admission ≥ 4.25 × 10^9^/L (OR 2.03 (1.31–3.16) *p* = 0.002), CRP at admission ≥ 59.5 (mg/dL) (OR 1.85 (1.18–2.89) *p* = 0.007), ferritin at admission ≥ 589 (ng/mL) (OR 1.66 (1.03–2.66) *p* = 0.04), Δ eosinophils ≥ 0.05 (OR 2.03 (1.30–3.17) *p* = 0.002), and oncological diseases (OR 2.44 (1.32–4.51) p =0.04) were predictors of persistent radiological abnormalities at three months after discharge (Table 2). In the multivariable analysis, age ≥ 62 years (OR 1.76 (1.05–2.8) *p* = 0.03) and Δ eosinophils ≥ 0.05 (OR 1.75 (1.05–2.9) *p* = 0.03) were the two independent predictors of residual lung abnormalities in the whole patient population (Figure 2). It is worth noting that, in the overall population, only 21 patients were reported as asthmatic, equally distributed between Rec (11 patients) and Not-Rec (10 patients).

## 4. Discussion

In this study, we evaluated the role of the eosinophil count in the course of COVID-19 pneumonia and the persistence of residual lung abnormalities in chest CT scans three months after discharge. We found that a lower eosinophil count is associated with more severe pneumonia. At the same time, a significant increase during the course of the disease resulted in an independent predictor for the persistence of residual lung abnormalities. In line with the literature, our study also confirmed that NLR, D-Dimer, CRP, and ferritin were significantly higher in patients with severe COVID-19 [9,10].

Despite a general initial eosinopenia, we observed eosinophils rising in both COVID-19 severity groups, and unexpectedly, this increase was higher in HIMC patients. Even though eosinophils are frequently associated with Th2 responses, their granules contain main Th1 cytokines such as interleukin (IL)-2, IL-12, and interferon-gamma [11,12,13]. Thus, beyond playing an important role in fighting parasites, eosinophils are also involved in the defense against bacteria, fungi, and viruses such as SARS-CoV2 [14,15,16,17]. Intriguingly, in a study by Fraiss’e et al., unpredicted eosinophilia was reported as a late-onset event during an intensive care unit stay, showing a positive impact on survival [18]. Eosinophil recovery seems essential in predicting the disease course, and persistent eosinopenia might be an ominous sign of severe illness with a higher risk of death [19,20,21,22]. Survivor bias may explain the increase in eosinophil count in HIMC patients due to the inclusion, for the purposes of this study, of patients with a three-month follow-up after hospital discharge. Conversely, this extra increase may have induced an overprotective repair process and, consequently, a higher prevalence of residual lung abnormalities. Despite their protective role in COVID-19 and other infectious diseases (such as influenza or parasitic infection), eosinophils and Th2 cytokines (especially IL-4 and IL-13) were repeatedly reported in the pathogenesis of idiopathic pulmonary fibrosis [23] and in other lung diseases, suggesting a possible role in the aberrant wound healing process. Moreover, IL-5 can promote fibrosis in the lung by recruiting eosinophils that produce transforming growth factor-beta1, platelet-derived growth factor, and IL-13 [24]. Yang Z.L. et al. showed a higher level of eosinophil count in COVID-19 patients, with evidence of fibrotic change during the radiological follow-up [25]. Toraldo and coworkers also reported the same result in their study; the eosinophil count, IL-6, and glutamic pyruvic transaminase were significantly associated with radiological residual abnormalities at three months [26]. Other authors suggest that eosinophils play a role in releasing pro-fibrotic cytokines in severe COVID-19 [27]. Some reports that long predate the pandemic have suggested that eosinophil recruitment on the lung in the bleomycin model can induce inflammation and, consequently, fibrosis [28]. In an in vitro model reported by Shock et al., a conditioned medium obtained by cultured eosinophils was demonstrated to be a strong stimulator for human fibroblasts. Moreover, they suggested that eosinophils can both adhere to and release mitogens for fibroblasts in vitro [29].

In 2022, Nguyen and coworkers also speculated that, in acute exacerbation of pulmonary fibrosis, eosinophils could have a central role and be a future target for new therapies [30]. Acute exacerbation of pulmonary fibrosis is a severe complication associated with high mortality. The specific mechanism is not fully understood [31]. However, some authors have shown that baseline cardiovascular diseases, a higher stage (≥II), and a higher eosinophil percentage (≥3.21%) in bronchoalveolar lavage fluid samples were predictors of an acute exacerbation of idiopathic pulmonary fibrosis [32].

The role of eosinophils in the development of long COVID has also been a recent topic of research. Of interest, in a study conducted in 2022 by Jukema, B.N., et al., the authors suggest that the neutrophil and eosinophil compartments are affected by COVID-19 in the long term and may be involved in the pathogenesis of long COVID. In particular, although blood eosinophils were lower during the phase of acute infection of COVID-19, their numbers did not fully normalize, with continuous hyper-responsiveness and prolonged activation in the eosinophilic compartment up to 6 months after the active phase of the disease [33]. The same suggestion was also reported in another study by Costa et al. In patients with COVID-19 and eosinophilia, eosinophils also increased after 90 days of the acute phase, with normalization at six months [34]. More studies are needed to understand the long-term effects of COVID-19 and granulocyte populations’ role in immune activation. It is essential to identify markers for patients at risk of developing a more severe presentation of long-term COVID-19 symptoms and to enable better management of this important condition, which seems to affect many people worldwide [35].

In our study, by multivariable analysis, we showed that a greater increase in eosinophils during hospitalization and older age were two independent risk factors for residual lung abnormalities at the three-month follow-up. Excessive eosinophil recruitment may indirectly affect tissue remodeling and repair, especially in older people. Despite their protective role against multiple organisms, such as parasites, viruses, and fungi, in a specific subgroup of frail and older patients, the higher activation and, consequently, intense production of fibrotic cytokines can promote the development of pulmonary sequelae. Further studies are needed to validate our hypothesis.

Our study has some limitations: first, as an observational, retrospective, and monocentric study, our results are only valid in our population; larger populations are needed to confirm the results. However, every effort was made to reduce missing data and transcription errors. Second, as stated, the inclusion criteria may have created survivor bias by excluding all patients not having a three-month radiological follow-up; however, this may be seen as a strong point in evaluating eosinophils as a marker to predict residual lung abnormalities. Further studies are needed to evaluate a eosinophil increase as a predictor of mortality in patients with severe COVID-19. Third, the variability of the blood count was only evaluated at two time points, admission and discharge, so this study cannot explain the role of eosinophil variability during the hospital stay. Fourth, the medical therapies of the two groups were remarkably different, corresponding to the intensity of the cure needed. The impact of medical therapy on eosinophils is something that we cannot fully elucidate. Finally, we do not have other data regarding a longer follow-up to report, but this is our goal in the future.

## 5. Conclusions

In conclusion, our preliminary results showed that a eosinophil increase during hospitalization could be a potential predictor of pulmonary sequelae in surviving patients after COVID-19 pneumonia. Despite the protective role of eosinophils, their excessive activation can be associated with radiological sequelae after the acute phase of SARS-CoV2 infection. More extensive studies are needed to confirm our results.

## Figures and Tables

**Figure 1 diagnostics-14-02320-f001:**
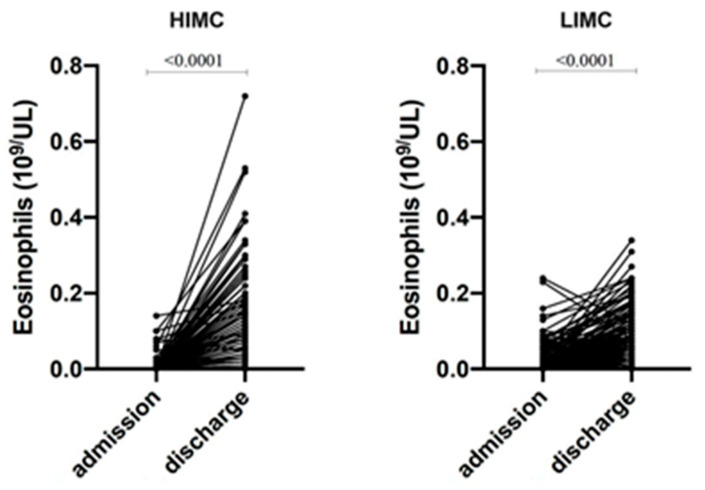
Eosinophil trends from admission to hospital discharge in HIMC and LIMC groups. Wilcoxon signed-rank test was used to compare eosinophil values between admission and discharge in HIMC patients (*p* < 0.0001) and LIMC patients (*p* < 0.0001). HIMC: high-intensity medical care; LIMC: low-intensity medical care.

**Figure 2 diagnostics-14-02320-f002:**
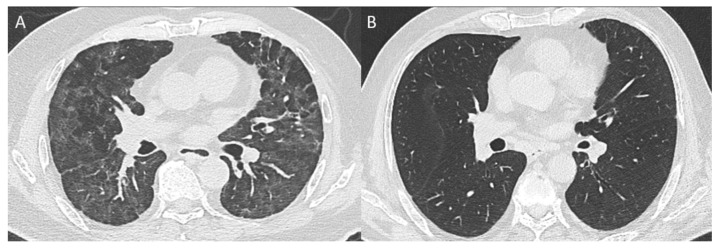
Example of chest CT scan 3 months after hospital discharge. Panel (**A**): patient with HIMC and a high rise in eosinophil count with GGO and reticular lung abnormalities (Not-Rec); panel (**B**): patient with HIMC and a low rise in eosinophil count without residual lung abnormalities (Rec).

**Table 1 diagnostics-14-02320-t001:** Demographics and clinical characteristics of the overall population and patients divided into low-intensity medical care group (*n* = 214) and high-intensity medical care group (*n* = 113).

	Overall (*n* = 327)	LIMC (*n* = 214)	HIMC (*n* = 113)	*p*
Age (years)	62 (22–88)	60 (22–87)	65 (25–88)	**0.01**
Sex—male, n° (%)	210 (64%)	137 (64%)	73 (65%)	0.92
BMI (kg/m^2^)	27 (16–57)	26 (16–57)	27 (19–48)	0.19
Pack-years—n°	10 (0.5–90)	11 (1–66)	10 (0.5–90)	0.78
Smoking History—n°	126 (39%)	75 (35%)	51 (45%)	0.09
Comorbidities				
Cardiological—n°	162 (50%)	92 (43%)	70 (62%)	**0.001**
Pneumological—n°	51 (16%)	37 (21%)	14 (12%)	0.24
Immunological—n°	45 (14%)	29 (16%)	16 (16%)	0.88
Metabolic—n°	151 (46%)	85 (40%)	66 (58%)	**0.001**
Oncological—n°	54 (17%)	31 (14%)	23 (20%)	0.17
Symptoms at admission				
Fever—n°	303 (93%)	194 (91%)	109 (96%)	0.06
Asthenia—n°	117 (36%)	81 (38%)	36 (32%)	0.28
Dyspnea—n°	156 (48%)	84 (39%)	72 (64%)	**0.0003**
Cough—n°	184 (56%)	120 (56%)	64 (57%)	0.92
GI symptoms—n°	71 (22%)	44 (21%)	27 (24%)	0.39
P/F	283 (40–542)	309 (121–542)	224 (40–461)	**<0.0001**
FiO_2_ at admission	21 (21–100)	21 (21–88)	29 (21–100)	**<0.0001**
Hospitalization—days	11 (2–67)	8 (2–49)	18 (3–67)	**<0.0001**
CT scan at admission				
- Alveolar score—%	7 (0–62)	5 (0–38)	18 (0–62)	**0.01**
- Consolidation score—%	1 (0–26)	0.8 (0–10)	4 (0–26)	**0.001**
- Interstitial score—%	1.6 (0–29)	0.8 (0–29)	10 (0–23)	**0.001**
Not-Rec patients—n°	159 (49%)	87 (41%)	72 (64%)	**0.0007**
∆ eosinophils	0.05 (−0.15–0.72)	0.04 (−0.15–0.3)	0.1 (−0.03–0.72)	**<0.0001**

BMI: body mass index, CT scan: computer tomography, Not-Rec: Not Recovered. Values are expressed as numbers and (%) or median and range, as appropriate. To compare demographics between HIMC and LIMC, the chi-square test and Fisher’s *t*-test for categorical variables and Mann–Whitney *t*-test for continuous variables were used.

**Table 2 diagnostics-14-02320-t002:** Predictive factors of radiological sequelae at first follow-up in all patients hospitalized for SARS-CoV-2-related pneumonia (HIMC + LIMC).

	Univariable		Multivariable	
	OD (0.95CI)	*p*	OD (0.95Cl)	*p*
Age ≥ 62 Years	2.04 (1.3–3.2)	**0.002**	1.75 (1.05–2.94)	**0.03**
Sex—Male	1.37 (0.87–2.16)	0.17	-	-
BMI ≥ 27 (kg/m^2^)	0.95 (0.59–1.50)	0.82	-	-
Severity—HIMC (yes)	2.56 (1.60–4.10)	**0.0001**	1.53 (0.86–2.72)	0.15
Pre-Admission Hematological Values				
LM Ratio ≥ 2.16	1.23 (0.79–1.91)	0.35	-	-
NL Ratio ≥ 4.64	1.66 (1.07–2.58)	**0.02**	0.87 (0.47–1.62)	0.66
Neutrophils ≥ 4.25 (×10^9^/L)	2.03 (1.31–3.16)	**0.002**	1.40 (0.78–2.53)	0.26
Lymphocytes ≥ 0.92 (×10^9^/L)	0.89 (0.58–1.39)	0.62	-	-
Monocytes ≥ 0.43 (×10^9^/L)	0.87 (0.57–1.35)	0.55	-	-
Eosinophils ≥ 0 (×10^9^/L)	0.78 (0.49–1.23)	0.28	-	-
CRP ≥ 59.5 (mg/dL)	1.85 (1.18–2.89)	**0.007**	1.04 (0.57–1.88)	0.89
D-Dimer ≥ 169 (mcg/mL)	1.21 (0.77–1.90)	0.42	-	-
Ferritin ≥ 589 (ng/mL)	1.66 (1.03–2.66)	**0.04**	1.50 (0.84–2.49)	0.18
Pre-Discharge Hematological Values				
LM Ratio ≥ 2.63	1.2 (0.78–1.86)	0.40	-	**-**
NL Ratio ≥ 2.45	0.92 (0.59–1.42)	0.70	-	**-**
Neutrophils ≥ 4.58 (×10^9^/L)	1.36 (0.88–2.10)	0.17	-	**-**
Lymphocytes ≥ 1.17 (×10^9^/L)	1.39 (0.90–2.16)	0.13	-	**-**
Monocytes ≥ 0.66 (×10^9^/L)	1.54 (0.99–2.39)	0.051	-	**-**
Δ Eosinophils ≥ 0.05	2.03(1.30–3.17)	**0.002**	1.75 (1.05–2.9)	**0.03**
CRP ≥ 6.00 (mg/dl)	0.92 (0.58–1.46)	0.72	-	**-**
D-Dimer ≥ 191 (mcg/mL)	1.27 (0.73–2.2)	0.39	-	**-**
Ferritin ≥ 723 (ng/mL)	2.25 (0.68–7.41)	0.18	-	**-**
Cardiological—yes	1.29 (0.84–1.99)	0.25	-	**-**
Oncological—yes	2.44 (1.32–4.51)	**0.04**	1.8 (0.9–3.7)	0.09
Pneumological—yes	1.12 (0.61–2.03)	0.71	-	**-**
Metabolic—yes	1.03 (0.67–1.59)	0.89	-	**-**
Autoimmunity—yes	1.53 (0.81–2.89)	0.19	-	**-**

BMI: body mass index, LM: lymphocyte–monocyte ratio, NL: neutrophil-to-lymphocyte ratio, CRP: C-reactive protein, HIMC: high-intensity medical care, OD: Odds ratio, CI: confidence interval.

## Data Availability

Dataset will be made available upon reasonable request to the corresponding author.

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
