# Peer review of "Blood Eosinophils Matter in Post-COVID-19 Pneumonia"

_diagnostics, 2024, doi:10.3390/diagnostics14202320_

Round 1

Reviewer 1 Report

Comments and Suggestions for Authors

This is a very interesting study investigating the role of eosinophils in COVID-19 pneumonia and the subsequent pulmonary sequela, with a major clinical importance, as the peripheral blood eosinophil count is highlighted as a potential biomarker for the identification of vulnerable individuals with a high probability to develop post-COVID-19 lung disease. The authors are advised to comment in the Results section on whether some of the patients had comorbidities with a predominance of eosinophilic inflammation, such as asthma, and how this could have affected the phenomenon of the eosinophil increase during hospitalisation. As a minor comment, the authors are advised to recheck the values of pack-years in Table 1 (they are reported as zero "0" for all patient groups).

Comments on the Quality of English Language

The language used is generally of good quality. As minor comments, the authors are advised to recheck all abbreviations (some of them are not explained in the text or appear only once, and it is thus unnecessary to be abbreviated), the American vs. British style (e.g. "dyspnoea" in Table 1 and "dyspnea" in the text), and the unnecessary capitalisation of some words (e.g. "Computed Tomography" in the Abstract or "Ferritin" in the Introduction).

Author Response

Comment 1

This is a very interesting study investigating the role of eosinophils in COVID-19 pneumonia and the subsequent pulmonary sequela, with a major clinical importance, as the peripheral blood eosinophil count is highlighted as a potential biomarker for the identification of vulnerable individuals with a high probability to develop post-COVID-19 lung disease. The authors are advised to comment in the Results section on whether some of the patients had comorbidities with a predominance of eosinophilic inflammation, such as asthma, and how this could have affected the phenomenon of the eosinophil increase during hospitalisation. 

We want to thank the reviewer for the kind comments and the help in improving our paper. We added the requested data about asthmatic/hypereosinophilic patients (line 160), however only 21 (6%) patients presented asthma, and no-one other hypereosinophilic diseases. The 21 patients were equally distributed (10 in the not-recovered and 11 in the recovered population)

Comment2

As a minor comment, the authors are advised to recheck the values of pack-years in Table 1 (they are reported as zero "0" for all patient groups).

We greatly appreciate the reviewer for pointing out this mistake, we corrected the wrong line in the table with the right values.

Comment 3

The language used is generally of good quality. As minor comments, the authors are advised to recheck all abbreviations (some of them are not explained in the text or appear only once, and it is thus unnecessary to be abbreviated), the American vs. British style (e.g. "dyspnoea" in Table 1 and "dyspnea" in the text), and the unnecessary capitalisation of some words (e.g. "Computed Tomography" in the Abstract or "Ferritin" in the Introduction).

We apologise for all the typos. We now amended the manuscript to avoid typos and tried to improve editing accordingly.

Reviewer 2 Report

Comments and Suggestions for Authors

Many thanks for the opportunity to review this study that I find interesting. In this study authors showed how eosinophil count might have a role as a predictor of disease severity and lung abnormalities on chest CT scans after 3 months. I would suggest that authors extend the introduction describing the relationship between eosinophil count and COVID-19 pneumonia based on the current litterature. In the method should be specified the period of time of the pandemic wave included in this study and the COVID-19 variants. Authors in the methods and results should also specify if they included breakthrough COVID 19 infection with pneumonia (pneumonia in vaccinated). Authors should specify in a Table the residual lung abnormalities found in the CT control after 3 months. In the discussion, authors should enhance how the eosinophils could have a role in the long COVID

Author Response

Many thanks for the opportunity to review this study that I find interesting. In this study authors showed how eosinophil count might have a role as a predictor of disease severity and lung abnormalities on chest CT scans after 3 months.

We thank the reviewer for his effort in reviewing our paper.

Comment 1 

I would suggest that authors extend the introduction describing the relationship between eosinophil count and COVID-19 pneumonia based on the current litterature. 

We thank the reviewer for this very appropriate comment. We extended the introduction as requested (please see line 34-40).

Comment 2

In the method should be specified the period of time of the pandemic wave included in this study and the COVID-19 variants. 

We thank the reviewer for this advice which helps us to properly present our population. We added this information in the methods section, as requested (please see line 46-50).

Comment 3

Authors in the methods and results should also specify if they included breakthrough COVID 19 infection with pneumonia (pneumonia in vaccinated). 

We thank the reviewer for pointing out the possibility of breakthrough infections, however, none of the patients was vaccinated due to the early enrollment in our population. We explained that nobody was vaccinated in the results section (line 99).

Comment 4

Authors should specify in a Table the residual lung abnormalities found in the CT control after 3 months. 

We really thank the reviewer for this comment. Unfortunately, we haven't scored yet (in terms of ground glass, bronchiectasis, etc..) all the CT scans after 3 months, as we want to show only our preliminary results. We only dichotomized the presence of lung abnormalities as “yes” or “no” (please see the method section).

Comment 5

In the discussion, authors should enhance how the eosinophils could have a role in the long COVID.

We agree that the paragraph dedicated to the role of eosinophils in long-COVID was not correctly described in the manuscript. Therefore, following the suggestion we amended the manuscript in the discussion. 

Reviewer 3 Report

Comments and Suggestions for Authors

good correlation. although i do not agree with the findings but the authors have shown statistically significant association in this population. may be it can be added that the results are valid in the study population only.

Author Response

good correlation. although i do not agree with the findings but the authors have shown statistically significant association in this population. may be it can be added that the results are valid in the study population only.

We thank the review for this comment. We absolutely agree that we should underline the need to confirm our findings in a larger population. Therefore, we modified both the limitations and conclusion to better explain this correct point (line 238-240).

Round 2

Reviewer 2 Report

Comments and Suggestions for Authors

Many thanks for the adjustments